# Mediastinal Parathyroid Cancer

**DOI:** 10.3390/cancers14235852

**Published:** 2022-11-28

**Authors:** Klaus-Martin Schulte, Gabriele Galatá, Nadia Talat

**Affiliations:** 1Academic Department of Surgery, School of Medicine and Psychology, College of Health and Medicine, Australian National University, Canberra, ACT 2601, Australia; 2Department of Endocrine Surgery, King’s College Hospital Foundation Trust, London SE5 9RS, UK

**Keywords:** primary hyperparathyroidism, parathyroid adenoma, parathyroid carcinoma, mediastinum, ectopic parathyroid, surgery, oncological surgery, prediction

## Abstract

**Simple Summary:**

Mediastinal parathyroid neoplasms (MPN) are a rare cause of primary hyperparathyroidism (pHPT). Pre-operative recognition of potential malignancy is the best way to inform adequate surgical access and enable curative oncological surgery. Analysis of a large group of patients with MPN identifies clinical characteristics of mediastinal parathyroid cancer (MPC). We propose a simple composite indicator of size and hypercalcemia, the 3 + 3 rule, to accurately predict malignancy in the majority of patients with MPN.

**Abstract:**

Parathyroid cancer (PC) is rare, but its pre-operative recognition is important to choose appropriate access strategies and achieve oncological clearance. This study characterizes features of mediastinal parathyroid cancer (MPC) and explores criteria aiding in the pre-operative recognition of malignancy. We assembled data from 502 patients with mediastinal parathyroid neoplasms (MPNs) from a systematic review of the literature 1968–2020 (*n* = 467) and our own patient cohort (*n* = 35). Thirty-two of the 502 MPNs (6.4%) exhibited malignancy. Only 23% of MPC patients underwent oncological surgery. Local persistence and early recurrence at a median delay of 24 months were frequent (45.8%), and associated with a 21.7-fold (95%CI 1.3–351.4; *p* = 0.03) higher risk of death due to disease. MPCs (*n* = 30) were significantly larger than cervical PC (*n* = 330), at 54 ± 36 mm vs. 35 ± 18 mm (χ^2^ = 20; *p* < 0.0001), and larger than mediastinal parathyroid adenomas (MPA; *n* = 226) at 22 ± 15 mm (χ^2^ = 33; *p* < 0.01). MPC occurred more commonly in males (60%; *p* < 0.01), with higher calcium (*p* < 0.01) and parathyroid hormone (PTH) levels (*p* < 0.01) than MPA. Mediastinal lesions larger than 3.0 cm and associated with a corrected calcium ≥ 3.0 mM are associated with a more than 100-fold higher odds ratio of being malignant (OR 109.2; 95%CI 1.1–346; *p* < 0.05). The composite 3 + 3 criterion recognized 74% of all MPC with an accuracy of 83%. Inversely, no MPN presenting with a calcium < 3.0 mM and size < 3.0 cm was malignant. When faced with pHPT in mediastinal location, consideration of the 3 + 3 rule may trigger an oncological team approach based on simple, available criteria.

## 1. Introduction

Primary hyperparathyroidism (pHPT) is moderately common, affecting up to 1% of the elderly population. The vast majority of cases, i.e., >98%, relate to mostly solitary and occasionally multiple benign parathyroid neoplasms [1], of which a significant fraction can be treated conservatively [2], unless bone and renal complications or significant hypercalcemia mandate surgery [1,3]. Due to complex embryological routing [4,5,6], ectopic parathyroids are not uncommon. Mediastinal locations, defined as inferior of the incisura jugularis and the upper chest aperture, account for approximately 1–2% of all cases of pHPT [6]. Whilst undescended glands pose challenges in the lateral neck compartment, thoracic descent results in mediastinal parathyroid neoplasms (MPN), eventually located as deep in the chest as the aortopulmonary window [7,8]. Such ectopic glands pose challenges to pre-operative localization [9] and are a prominent driver of surgical failure [10]. Half a century later, a statement from pioneers at the Massachusetts General Hospital may still hold true: “Mediastinal parathyroid tumors in particular, have frustrated many surgeons” [11].

Parathyroid carcinoma (PC) is a clinically and genetically complex condition in want of better care [12,13,14,15,16]. Parathyroid carcinoma accounts for merely 0.05‰ of all cancers reported in the NCBD database [17], and 1–2% of all cases of primary hyperparathyroidism (pHPT) [18]. The prognosis of PC remains guarded, with a 5-year survival rate of only 80% [19], depending on the radicality of the initial surgery. Patients who undergo parathyroidectomy have a remarkably better disease specific survival (DSS) and overall survival (OS) than patients who do not undergo definitive treatment [20] and more extensive oncological surgical approaches produce better results [20], in line with recommendations to aim for margin free outcomes in the first operation [21,22]. Failure at the initial stage is difficult to recuperate [23,24].

It is self-evident that lesions deep in the thoracic cavity particularly challenge oncological clearance due to access problems, adjacent vital structures, and a lacking overlap or coordination of endocrine and thoracic-mediastinal surgical expertise, even in centers of excellence. Nation-wide cohort studies support the claim that higher surgeon annual volume is associated with decreased rates of repeat parathyroid surgery in benign circumstances [25]. Upfront referral to expert centers might significantly improve the prospects of patients suffering from PC.

The best attempts to funnel patients into optimal care pathways start with the recognition of potential malignancy. At present, there is a dearth of criteria to identify parathyroid cancer before surgery and histological examination. Potential malignancy is indicated by a syndromic context such as Hyperparathyroidism-Jaw Tumour Syndrome (HP-JTS) with CDC73 mutations [26,27]. Cross-sectional imaging [28] and MIBI SPECT-CT [29,30,31] have limited differentiating power. The contribution of FDG-PET CT is insufficiently explored [32], but F-18 fluorocholine PET/CT holds promise [33,34,35] and may detect lesions evading FDG-PET detection [33]. Some authors have demonstrated ultrasonographic features suggesting possible malignancy in parathyroid lesions > 15 mm [36], but the technique is not applicable to intrathoracic lesions except for the most superficial ones. Schulte et al. have proposed a composite criterion, the 3 + 3 rule, to clinically assess the risk of PC in cervical parathyroid lesions [12,37]. In a large cohort of patients with pHPT, they identified that lesions smaller than 3 cm and with a corrected serum calcium level below 3.0 mM carried next to no risk of malignancy, whilst lesions meeting both criteria had a malignancy risk in the order of 5–10% [12].

The particular constraints on surgical access to the mediastinum, the required subspecialty expertise, demands on appropriate planning for oncological surgery, the suspected poor outcomes of failed surgery in the mediastinum, and the need for realistic risk estimates to adequately involve patients in decision making and consent, have prompted us to explore the applicability of the 3 + 3 rule for the pre-operative recognition of potential malignancy.

## 2. Materials and Methods

### 2.1. Patients and Methods

This study comprises a novel cohort of 502 patients with mediastinal parahyroid neoplasm (MPN). For comparison purposes, we employed a cohort of 546 patients with cervical parathyroid neoplasms (CPN), comprising of 330 prior published patients with cervical parathyroid cancer (CPC) [38] and 216 patients with cervical parathyroid adenomas (CPA) operated in our service. The MPN cohort (*n* = 502) comprises a benign MPA cohort (*n* = 470) and a malignant MPC cohort (*n* = 32). The final MPA cohort of 470 patients includes 436 cases retrieved from the literature, and 34 patients who had been operated at our center. The final MPC cohort of 32 patients comprised of 31 cases retrieved from the literature, and 1 patient who had been operated at our center.

Patients from the literature were retrieved as follows: The NIH database PubMed (http://www.ncbi.nlm.nih.gov/sites/entrez; accessed on 30 June 2022) was searched using the search terms ‘ectopic’ and ‘parathyroid’ and ‘cancer’ or ‘carcinoma’. The search identified 786 articles. The PRISM diagram (Figure 1a) provides detail. The NIH database PubMed (http://www.ncbi.nlm.nih.gov/sites/entrez, accessed on 30 June 2022) was also searched using the search terms “parathyroid ectopic” and “parathyroid mediastinal” to identify mediastinal parathyroid adenomas. The search identified 2337 publications. The PRISM diagram (Figure 1b and Appendix A) provides detail.

### 2.2. Eligibility Criteria

Articles published in English, French, German and Spanish were included. After review of the title, abstract and keywords, 31 cases of parathyroid cancer were identified from 29 publications published between 1968 and June 2022 [39,40,41,42,43,44,45,46,47,48,49,50,51,52,53,54,55,56,57,58,59,60,61,62,63,64,65,66,67]. A total of 436 cases of mediastinal parathyroid adenomas were identified from 221 articles published between 1990 and 2020. A hand search of articles’ bibliography was also performed using the ISI Thompson Web of Knowledge Citation report to identify additional patients with MPC. Patients with secondary hyperparathyroidism were excluded. A Google Scholar search in June 2022 did not identify any further articles reporting on individual patients with MPC.

### 2.3. Data Selection

We identified surgical approaches as described by prior publication [68]. We used our prior-reported cohorts of cervical parathyroid carcinoma (PC) and cervical parathyroid adenoma (PA) [38] for comparative analysis with the newly established cohorts of mediastinal parathyroid adenoma (MPA) and carcinoma (MPC).

The methodology for PTH measurement has changed several times during the observation interval spanning the time between 1968 and 2000. We have, therefore, not provided any values for PTH levels, but calculated a dimensionless figure for PTH levels as “times upper normal limit” (xUNL) by dividing the reported PTH level in any study, regardless of units, by the reported value for the upper normal limit in that same study.

### 2.4. Definition of the 3 + 3 Rule

The 3 + 3 rule was established for cervical parathyroid lesions and aims to predict malignancy [12]. It is based on two simple criteria, i.e., lesion size ≥ 3.0 cm OR corrected serum calcium levels ≥ 3.0 mM or both. A lesion is considered to fulfill the 3 + 3 exclusion rule if it has a size < 3.0 cm AND presents with a corrected serum calcium level of <3.0 mM. Such lesions are likely benign.

### 2.5. Statistical Analysis

SPSS version 28 statistical program was used to calculate mean, median, chi-square values, relative risk, odds ratios and scatter plots. Definitions used for risk analysis [69] and parameters of assay quality are established.

## 3. Results

### 3.1. Clinical Characteristics and Outcomes in the MPC Cohort

The MPC cohort comprised 32 patients [39,40,41,42,43,44,45,46,47,48,49,50,51,52,53,54,55,56,57,58,58,59,60,61,63,64,65,66,67,70]. Additional detail on individual patients is provided in Appendix A. Age at diagnosis was known for 30 patients, and a mean age of 54.8 ± 16.4 years, with a median of 56.5 and a range of 10–84 years, was reported. Eighteen patients (60%) were male and 12 female (40%), calculating as a gender ratio of 1.5. The corrected serum calcium level was known for 25 patients, and was 3.6 ± 0.7 mmol/L, with a median of 3.6 and a range of 1.9–5.8 mmol/L. Preoperative PTH levels were known for 23 patients, and reported to 16.8 ± 12.8 above upper normal limit (UNL), with a median of 16.9 UNL, and a range of 2.1–47.8 UNL. Defined at a corrected serum calcium level ≥3.0 mM and a PTH level ≥ 10-fold the upper normal limit, marked excess was present in 85.2% and 65.2%, respectively. Lesion size was reported in 24 patients, with a mean of 5.48 ± 3.7 cm, a median of 4.25 cm and a range of 2.0–15.5 cm. Lesion size was ≥3.0 cm in all but two cases (92.0%). Surgery involved local excision (LE) in 20 cases (62.5%), and en bloc (EB) resection in 7 cases (21.8%). Lymph nodes status was reported in only 7 patients (21.9%), with nodal metastases identified in 3 (9.4%) and no nodes in 4 cases (12.5%). Margin status was known only in our own patient. In 24 patients with known outcome, 5 persistences (20.8%) were reported, whilst in 19 patients with known outcome, 6 recurrences (31.6%) were reported with a delay of 22.3 ± 15.2 months, at a median of 24 and range of 0–84 months. Together, 11 patients (45.8%) suffered recurrence or persistence, all of which included local cancer manifestations. Of these patients, 8 (72.7%) had only local disease, whilst two 18.2% were identified to also include distant metastases. Follow-up (FU) data were reported in 21 patients, with a mean FU of 31.6 ± 24.7 months, a median of 24 and a range of 1–84 months. Six of these 21 patients (28.6%) were reported to have died of disease (DOD) during this FU interval. Cancer specific mortality was exclusively linked to locoregional persistence or recurrence (6/6 cases, 100%). Failure to achieve locoregional clearance, indicated by local persistence or recurrence (*n* = 11), carried a 21.7-fold (95%CI 1.3–351.4; *p* = 0.03) relative risk of death due to disease, with an Odds Ratio OR of 46.1 (95%CI 2.2–952; *p* = 0.01).

### 3.2. Comparative Analysis

Comparative analysis showed MPC to be significantly larger in size than cervical PC (mean 34.5 ± 17.5 mm vs. 54.8 ± 37.0 mm, *p* < 0.0001). Patients with MPC were significantly older than those with cervical PC at 54.8 ± 16.4 years and 48.5 ± 15.0 years, respectively (*p* < 0.05). Gender, PTH and corrected calcium levels did not differ significantly between the two cohorts (Table 1).

MPCs were significantly larger in size than MPAs (mean 54.8 ± 37.0 mm vs. 22.1 ± 14.5 mm, *p* < 0.001) (Table 1). Gender, PTH and corrected calcium levels also differed significantly between the two cohorts (Table 1).

### 3.3. Segregation of Parathyroid Cancer Status with a Composite Criterion of Size and Degree of Hypercalcemia

When plotting lesion size and corrected serum calcium levels prior to the first surgical intervention, benign and malignant cervical lesions segregate along the 3 + 3 criteria as prior demonstrated [12] (Figure 2a). MPC segregated in the same fashion as cervical PC (Figure 2a). A similar near-complete segregation was observed between benign MPN from patients treated in our own service and MPC (Figure 2b) and all benign MPN identified through the literature search (Figure 3b). For comparison: mediastinal parathyroid adenomas are generally associated with higher calcium levels, but not larger size, than their cervical counterparts (Figure 3a).

### 3.4. The Composite Criterion of Size and Hypercalcemia Identifies the Malignancy Risk of MPN

We analysed 257 MPAs collected from our own patient cohort, small case series and individual case reports. Half of the cases (50.2%) fulfilled the 3 + 3 exclusion rule, defined as a lesion size < 3.0 cm concomitant with a corrected serum calcium level of <3.0 mM [12], whilst 16.0% were positive for both composite criteria, i.e., fulfilled the 3 + 3 rule (Table 2).

We then analyzed the impact of lesion size and hypercalcemia, defined at the thresholds of the 3 + 3 rule, on the risk of malignancy in the cohort comprising all mediastinal parathyroid neoplasms (MPNs). MPNs exhibiting a large size and hypercalcemia above threshold carried a much higher risk of malignancy (Table 3).

### 3.5. The Composite Criterion of Size and Hypercalcemia Identifies Malignancy with Resonable Sensitivity and Accuracy

MPNs exhibiting both, a size < 3 cm and a corrected serum calcium < 3.0 mM exclusively clustered in the benign cohort. Pre-operative presence of a large hypercalcemic mass identified MPC with a sensitivity of 73.9% and a specificity of 84.1%. The positive predictive value was 29.3%, meaning that nearly a third of lesions fulfilling both criteria prior to surgery were actual MPCs (Table 4).

Inversely, the 3 + 3 rule could also be used to exclude cancer, then used as 3 + 3 exclusion rule. Taking small size and moderate hypercalcemia as input, and the absence of MPC as output, the test has a specificity and positive predictive value of 100%. However, overall test performance is poor using these settings (Appendix A).

## 4. Discussion

Our findings are novel and relevant for our understanding of mediastinal parathyroid neoplasms: Mediastinal PC presents at an older age and with larger size than cervical PC, and is significantly larger and more hypercalcemic than its benign mediastinal counterpart, the mediastinal parathyroid adenoma. Mediastinal lesions smaller than 3 cm and with moderate hypercalcemia, i.e., <3.0 mM, are virtually never malignant whilst lesions larger than 3 cm and with excessive hypercalcemia (>3.0 mM) carry an about 100-fold higher risk of malignancy. Nearly one third of large (≥3 cm) mediastinal lesions with excessively hypercalcemia (≥3 mM) are expected to represent parathyroid cancer. These observations are highly useful to steer an upfront surgical approach aiming at free resection margins to grant best long-term outcomes.

It is hard to strike the right balance between oncological purpose and the avoidance of undue harm when faced with uncertainty whetherthe surgical target is benign or malignant. Often, it is advised that the surgeon must be attuned to intraoperative findings [13], implying a resection strategy based on ad hoc decision-making. In the absence of reliable indicators of malignancy, current guidelines recommend considering a diagnosis of PC when PTH is markedly elevated and hypercalcemia severe (strong recommendation; low-quality evidence) [22]. They advise considering the intraoperative suspicion of parathyroid carcinoma, then proceeding to complete resection avoiding capsular disruption with eventual en bloc resection of adherent tissues to improve the likelihood of cure (strong recommendation; low-quality evidence) [22]. Even in the surgery of cervical cases of pHPT, such advice is poorly heeded. Indeed, the majority of patients from large database cohorts undergo mere local resection, i.e., the standard procedure employed for benign parathyroid neoplasms, with “en bloc” resections accounting for hardly 10% of the patient cohort later identified to exhibit malignancy [19]. Data from the National Cancer Data Base from 1985 to 2006 identify that two thirds of patients with PC (65.7%; *n* = 733) underwent incomplete tumour removal, including local tumor destruction, local tumor excision, simple/partial removal of tumor, debulking [15]. Our own analysis of 1036 PC patients from the literature identified that 53% had undergone local excision alone [38]. Contemporary data from Germany confirm that the use of en bloc resection techniques almost doubles, when PC is suspected preoperatively [71].

This core challenge of surgical adequacy is even more prominent in mediastinal lesions. A pre-operative suspicion will not be prompted by otherwise useful indicators, including palpation [68], ultrasound [36], or the sensitive identification of typical lymph nodes by ultrasound.

A larger size of parathyroid lesions favors their detection by sestamibi scans (MIBI) [72], whilst PTH and calcium excess do not [73] and lymph nodes often remain unapparent on such scans [29]. A detailed examination in limited-size PC cohorts show that a combination of US with CT and MIBI or CT with MIBI achieves a higher accuracy in localization of PC [30], but do such tests not convincingly aid in the diagnosis of malignancy unless expert and, as yet, unconfirmed criteria are applied [31]. FDG-PET-CT adds little value [32].

In absence of suitable indicators of malignancy, surgeons will consider practicalities of access to thoracic lesions rather than oncological principles. Many ectopic mediastinal lesions can be accessed and removed by a low standard cervical Kocher incision [74] or minimally invasive access techniques [75,76], including video-assisted thoracoscopy VATS [77,78,79,80], or robot-assisted approaches [81]. Different from the experience in neck surgery, such approaches disable a comprehensive intraoperative circumspection and palpation of the lesion site before tissue planes are opened. They hence remove the single most important criterion guiding ad hoc decision-making regarding oncological proceedings.

The data presented here bear testimony to this inability to correctly identify and treat parathyroid cancer in the mediastinum and further support the link between surgical adequacy, locoregional disease control and cancer-specific mortality, with both following initial surgery by a median delay of merely 2 years. With less than a quarter of patients (23.3%) undergoing oncological surgery, there is little evidence that MPC was in fact treated in line with general oncological standards: resection margins are not reported, except for a single case (96.7%) comparing to only a quarter in large database sets [19]. Lymph nodes were removed in merely 6.6% of patients, indicating lesser surgical performance than reported for cervical PC, where 35% of a nation-wide cohort of mostly cervical PC underwent some form of lymphadenectomy [19]. Accordingly, persistence or local recurrence are observed in 45.8% of patients with MPC. The initial failure of surgical cancer control is associated with a markedly (22-fold) higher risk of death due to cancer (*p* < 0.05).

In contrast to the current dearth of indicators in PC, our exploration of simple and available criteria to gauge the risk of malignancy prior to surgery has yielded useful outcomes. Information on both criteria will invariably be available: determination of corrected serum calcium levels is an integral component of the initial assessment of pHPT [82], whilst some form of cross-sectional imaging is a pre-requisite in the pre-operative localization foregoing surgery [73].

We demonstrate that MPC follows the clinical pattern observed in cervical PC (Figure 2a) [12]. Neither size nor the degree of hypercalcemia alone possess sufficient discriminatory power. This is due to the fact that mediastinal parathyroid adenomas exhibit considerable variability in size and calcium levels, overlapping with MPC (Table 1). However, and as in cervical PC, when both criteria are co-opted to yield a composite criterion called the 3 + 3 rule, excellent discrimination is achieved (Figure 2 and Figure 3). In MPC, the 3 + 3 rule works both ways. Small mediastinal lesions (<3 cm) with moderate hypercalcemia (<3.0 mM) have not been reported to represent MPC, with the “negative 3 + 3 rule” defining a low-risk scenario in MPN.

Taking this with a grain of salt, surgical removal can prioritize the practicalities of surgical access in such low-risk scenario. In absence of a prevailing oncological perspective, surgeons must acknowledge the comorbidities of patients which co-shape the outcome of any intrathoracic procedure. Independent of age, frailty indicates adverse outcomes after cardiac surgery [83]. However, systematic investigations into the outcomes of elective lung surgery show that the choice of procedure, i.e., video-assisted thoracic surgery VATS versus thoracotomy, does not impact morbidity, whilst the extent of the performed procedure persists as risk factor in multivariate analysis [84], findings likely owed to the complex pathophysiology of chest interventions [85]. While minimally invasive thoracic procedures doubtless carry a much lesser burden for the patients, they do not transform peri-operative morbidity and mortality, as those are driven by patient factors, rather than access choice.

On the other hand, the oncological perspective must gain the upper hand when the pre-operative risk of malignancy is high. Large mediastinal lesions (≥3 cm) with severe hypercalcemia (≥3.0 mM) are commonly malignant (29.3%, Table 3), the “positive 3 + 3 rule” defining a high-risk scenario in MPN. When faced with a positive outcome of the 3 + 3 rule, the odds of malignancy in any such lesion are 100-fold higher than in the absence of both criteria (*p* = 0.001).

The key limitations of this study are its retrospective nature, potential reporting bias, and the small sample size. The study cohort of MPC spans a wide age range from 10 to 84 years, with one non-adult patient, but is there little evidence that age would shape outcome expectations in parathyroid malignancy in a major way. These shortcomings are, at least in part, mitigated by the fact that mediastinal parathyroid neoplasms follow the segregation pattern according to the 3 + 3 rule observed in cervical parathyroid neoplasms. The observation that more than half of PC from large database studies are smaller than 3 cm [19] with a median range of tumour size of 2.6–3.0 cm [38] does not hinder the use of the 3 + 3 criterion. A vast proportion of PC cases are failed when the size criterion is used in isolation (Figure 2a), but their correct prediction is rescued by the co-presence of severe hypercalcemia (Figure 2a). MPC meets both 3 + 3 criteria even more commonly than cervical PC (Figure 3b). The observation that tumour size only variably correlates with long-term disease outcomes in PC [15,86,87,88] equally does not subtract from the value of our observations. The purpose of the 3 + 3 rule is not long-term outcome prediction, but the pre-operative identification of malignancy with the purpose of guiding an adequate surgical approach.

If treated correctly, PC clusters in a risk group of cancers with long-term survival in whom secondary malignancies, cardiac and other events far outweigh the risk of dying from PC [89]. In light of the poor outcomes of patients with recurrence or persistence of disease [13,38], the significant link between surgical under-treatment and poor outcomes (see [21]), and the significant technical complexities of oncological surgery in the mediastinum [90,91,92,93,94,95], we propose the following approach: unless individual observations indicate otherwise, patients with small lesions (<3.0 cm) and only moderate hypercalcemia (<3.0 mM) can be approached as if presenting with benign mediastinal adenomas. Patients with large lesions (≥3.0 cm) and concomitant severe hypercalcemia (≥3.0 mM) should be referred to centers with particular expertise in this to undergo planned oncological surgery. Patients with intermediate risk might also benefit from referral to centers.

## 5. Conclusions

As for cervical parathyroid neoplasms, the provision of the best surgical care in MPC requires the timely recognition of malignancy, which can be aided by application of the 3 + 3 rule. A positive outcome of the composite 3 + 3 criterion, i.e., the presence of a large lesion (≥3 cm) with concomitant severe hypercalcemia (≥3.0 mM), has a reasonable sensitivity (73.9%) and specificity (84.1%) to afford the pre-operative prediction of malignancy with an accuracy of 83.2%. About one third of such lesions (29.3%) will eventually be confirmed as malignant. Inversely, a negative outcome of the 3 + 3 rule for both criteria virtually excludes malignancy (PPV 100%). Patients meeting only one of the two criteria have an intermediate risk of malignancy (6.5%). Timely referral to centers with particular expertise may represent the current best option to improve outcomes in parathyroid cancer.

## Figures and Tables

**Figure 1 cancers-14-05852-f001:**
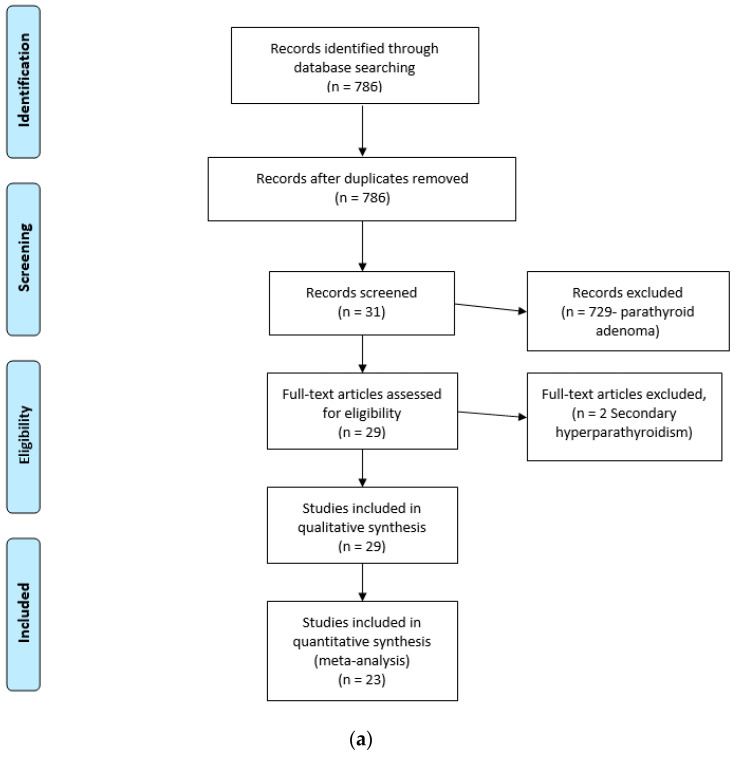
(**a**) PRISMA Flow Diagram for MPC. (**b**) PRISMA Flow Diagram for benign MPNs.

**Figure 2 cancers-14-05852-f002:**
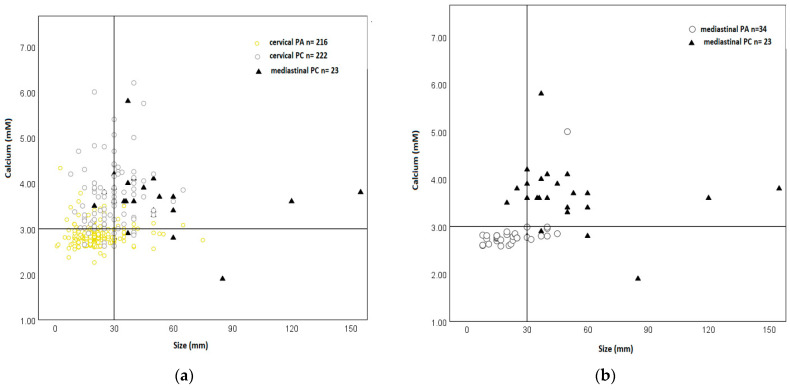
Scatter plot of calcium levels versus tumour size of benign and malignant parathyroid neoplasms in cervical (**a**) and mediastinal (**b**) location. Legend to Figure 1: The criterion for a lesion to be suspicious at a size > 3.0 cm or hypercalcaemia > 3.0 mM is referred to as the “3 + 3 rule” obtained from Schulte and Talat [12]. (**a**) (**left**) parathyroid adenomas and cervical PC have been obtained from Schulte and Talat [12]. (**b**) (**right**), the cohort of 23 mediastinal parathyroid cancer patients corresponds to the patients in Appendix A. The data for 34 mediastinal adenomas are from patients in our own service.

**Figure 3 cancers-14-05852-f003:**
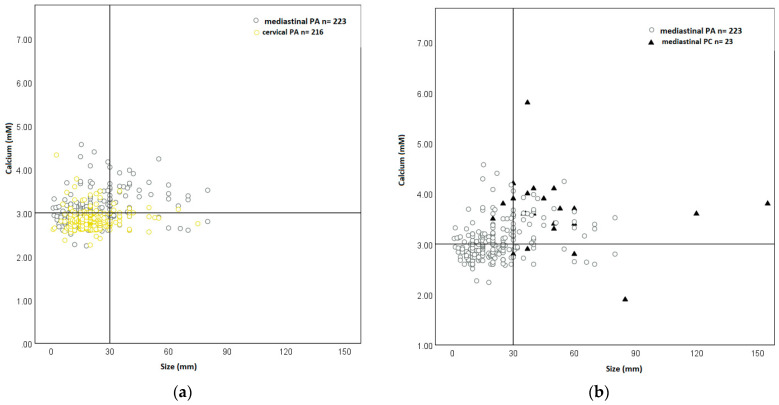
Scatter plot of calcium levels versus tumour size of benign (**a**) and malignant (**b**) mediastinal parathyroid neoplasms. Legend to (**a**) (**left**) cervical parathyroid adenomas been obtained from Schulte and Talat [12] and 223 mediastinal parathyroid adenomas were obtained from case reports from the literature. (**b**) (**right**), the cohort of 23 mediastinal parathyroid cancer patients corresponds to the patients in Appendix A.

**Table 1 cancers-14-05852-t001:** Clinical presentation of cervical and mediastinal parathyroid neoplasms.

		Cervical PC ^a^	Mediastinal PC	Chi sq (*p* Value)	Mediastinal PA	Mediastinal PC	Chi sq (*p* Value)
Age at diagnosis	Mean ± SD	48.5 ± 15.0	54.8 ± 16.4	4.3	51.1 ± 17.1	54.8 ± 16.4	n.s
	Median	48	56.5	*p* < 0.05	53	56.5	
	Range	13–84	10–84		8.0–88	10–84	
	Total with available data (*n*)	330	30		432	30	
Gender	(Male/Female)	151/179	18/12	n.s	127/299	18/12	11.8 *p* < 0.001
	Total with available data (*n*)	330	30		311	30	
Corrected calcium (mM)	Mean ± SD	3.6 ± 0.6	3.6 ± 0.7	n.s	3.0 ± 0.38	3.6 ± 0.7	6.1 *p* < 0.01
	Median	3.6	3.6		2.9	3.6	
	Range	2.6–6.2	1.9–5.8		2.2–5.0	1.9–5.8	
	Total with available data (*n*)	229	27		366	27	
PTH (xUNL)	Mean ± SD	11.6 ± 12.5	16.8 ± 12.8	n.s	7.0 ± 9.4	16.8 ± 12.8	6 *p* < 0.01
	Median	8	16.9		3.5	16.9	
	Range	1.0–30.0	2.1–47.8		1.0–90.0	2.1–47.8	
	Total with available data (*n*)	146	23		343	23	
Size of lesion (mm)	Mean ± SD	35.4 ± 17.5	53.5 ± 35.8	20.3 *p* < 0.0001	22.1 ± 14.5	53.5 ± 35.8	32.8 *p* < 0.001
	Median	34	42.5		20	42.5	
	Range	12–125	20–155		1–80	20–155	
	Total with available data (*n*)	222	26		336	26	

n.s.: not significant; ^a^: data of cervical parathyroid cancer was retrieved from previous publications (references [11,56]) to carry out comparative analysis. xUNL = times upper normal limit. The total cohort size is 32 patients with mediastinal cancer, yet complete information on all items was available for only 23 patients. In the mediastinal parathyroid cohort, 34 patients were from our center and 436 patients were from the literature.

**Table 2 cancers-14-05852-t002:** Lesion size and corrected calcium levels in mediastinal parathyroid adenoma (MPA).

All Mediastinal Parathyroid Adenomas (MPA)	<3 cm + <3 mM	≥3 cm + <3 mM	<3 cm + ≥3 mM	≥3 cm + ≥3 mM	Total
our cohort	25 (74%)	8 (24%)	0	1 (3%)	34
small case series	62 (61%)	4 (4%)	25 (25%)	10 (10%)	101
individual case reports	42 (34%)	11 (9%)	39 (32%)	30 (25%)	122
Σ of all MPAs	129 (50.2%)	23 (9.0%)	64 (24.9%)	41 (16.0%)	257

**Table 3 cancers-14-05852-t003:** Lesion size and corrected calcium define the cancer risk of in mediastinal parathyroid neoplasms (MPN).

coc	All MPN. *n* = 280	MPA *n* = 257	MPC *n* = 23	RR 95%CI Significance	OR 95%CI Significance
<3 cm + <3.0 mM	129	129	0	(control group)	(control group)
<3 cm + ≥3.0 mM	66	64	2	9.7 0.4–199.2 *p* = 0.14	10.4 0.5–219.1 *p* = 0.13
≥3 cm + <3.0 mM	27	23	4	41.7 2.3–754.2 *p* = 0.01	49.6 2.6–952.0 *p* = 0.009
either size ≥3 OR corr Ca ≥ 3.0 mM	93	87	6	18.0 1.0–315 *p* = 0.048	19.2 1.1–346 *p* = 0.045
≥3 cm + ≥3.0 mM	58	41	17	77.1 4.7–1261.0 *p* = 0.002	109.2 6.4–1856.0 *p* = 0.001

**Table 4 cancers-14-05852-t004:** The positive 3 + 3 criterion, large lesion and severe hypercalcemia, predicts presence of MPC. *p* = has cancer = 23; *n* = has no cancer = 257.

Cocal	Σ Test Pos	Sensitivity TP/P	Specificity TN/N	PPV TP/(TP + FP)	NPV TN/(TN + FN)	Accuracy (TP + TN)/(P + N)
all	280					
<3 cm + <3 mM (control)	129	0/23 0%	128/257 49.8%	0/129 0%	128/151 84.8%	128/280 45.7%
≥3 cm + <3 mM	27	4/23 17.4%	232/257 90.3%	4/27 14.8%	232/253 91.7%	236/280 84.3%
<3 cm + ≥3 mM	66	2/23 8.7%	193/257 75.1%	2/66 4.3%	193/214 90.2%	195/280 69.6%
≥3 cm or ≥3 mM	93	6/23 26.1%	170/257 66.1%	6/93 6.5%	170/187 90.9%	176/280 62.9%
≥3 cm + ≥3 mM	58	17/23 73.9%	216/257 84.1%	17/58 29.3%	216/222 97.3%	233/280 83.2%

## Data Availability

The authors confirm that the findings of this study are available within the article and its Appendix A. Raw data that support the findings of this study are available from the corresponding author, upon request.

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
