# Peer review of "Mediastinal Parathyroid Cancer"

_cancers, 2022, doi:10.3390/cancers14235852_

Round 1

Reviewer 1 Report (Previous Reviewer 1)

The Authors have properly revised the manuscript.

I have no more suggestions 

Author Response

Dear Reviewer,

thank you for your kind and speedy review. It is much appreciated.

With best wishes

Klaus-Martin Schulte 

Reviewer 2 Report (Previous Reviewer 3)

The author response properly to the question raised last time. No further question.

Author Response

Dear Reviewer,

thank you for your kind and efficient review. It is much appreciated.

With best wishes

Klaus-Martin Schulte

This manuscript is a resubmission of an earlier submission. The following is a list of the peer review reports and author responses from that submission.

Round 1

Reviewer 1 Report

The manuscript is interesting and originates from the idea to associate data from a systematic review of the literature  from 1968-2020 coupled to a patient cohort with mediastinal parathyroid neoplasms in order to better characterize mediastinal parathyroid carcinomas.

The authors applied the 3+3 criterion that indicates that mediastinal lesions larger than 3.0cm and associated with a corrected calcium ≥3.0 mM carries a more than 100fold higher odds of malignancy.

Though the manuscript has important key messages , it presents some criticisms

1 lines 53-54 : The prognosis of PC remains guarded, with a 5 year survival rate of only 80% [19].  –please specify that this  depends on the kind of first surgery in term of radicality .. please specify this point

2 Introduction should be shortened focusing on the topic of the manuscript

3 Patients and Methods  should be better presented…  the authors included 34 patients with MPA and one with MPC who had been operated at their  center in the analysis.   But previously they reported that they  have assembled patient data from a systematic review of the literature from 1968 up to 2020 and their own patient cohorts

4 From 1968 up to 2020 PTH measurement method likley changed: please specify this point

5 Tables should be shortened and graphically improved

6 The cohort exposes really a wide age range (10-84 years)  … form children to very old people .. this data can affect the results and should be seriously discussed

7 Discussion should be shortened and addressed to the differences between cervical and mediastinic PC  

8 a statistical revision should be encouraged 

Author Response

Dear Reviewer 1

thank you very much for the careful review of this manuscript which we have all integrated - see attached document. Thank you- we believe the manuscript is better for it!

With best wishes

Reviewer 2 Report

1.     Line 27 : Isn’t « odds » plural ?

2.     Lines 18, 19 : Please specify briefly : How many patients were from the “literature” and how many from “own cohorts”?

3.     Line 12: Please avoid claims like “largest ever/first ever/best ever” that the reviewers cannot verify with reasonable effort and that potentially can trigger objections from readers. Besides, strictly speaking, it was not “a” cohort, but an assembly of literature cases and the authors’ own cohort.

4.     Line 30: It is fine to raise awareness among surgeons, but why not among other specialists as well? As an endocrinologist, I also found the 3x3 “rule of thumb” very interesting and useful. Please do not underestimate the role of other specialists in educating, updating and guiding surgeons in optimizing their approach to endocrine tumors (benign and malignant). The authors should avoid giving the impression that the management of hyperparathyroidism should be surgeon-centered.

5.     Visual summary: Please specify “size” and “serum calcium”. Also, please remove the red triangle labelled “concern of malignancy”. First, it implies a linear increase in the risk, which may or may not be the case. Second, it is not necessary, because the data are shown above and the message is clear. Finally, as a minor issue, in the second pie chart, the red slice is longer than the green one, whereas in the pie chart on the right they are the same; please harmonize.

6.     Visual summary: The title is incorrect. Most lesions that fulfil the 3+3 criterion are benign. “Identifies” is too strong and unsupported by the data. The authors could say “increases the risk for”, “helps to identify”, or something similar.

7.     Line 40-42: Since you are referring to the criteria for surgery in cases of asymptomatic hyperparathyroidism, the age should also be mentioned. The most up-to-date reference to cite would be the following: https://doi.org/10.1002/jbmr.4677

8.     Line 43: Please provide in the text a clear definition for “mediastinal localization”. Specifically, state what is the upper limit for considering a parathyroid tumor’s location as mediastinal. Is it the upper limit of the manubrium?

9.     Line 52: Please confirm that you really mean 0.05 per thousand and not 0.05 per cent.

10.  Line 55: Please correct “case” to “cause”.

11.  Line 58: What does “(Schulte et al 2022 – this issue of Cancers)” refer to?

12.  Line 65: “Albeit this will likely never be demonstrated by high-level evidence”. This statement is not necessary and may undercut the message. I recommend deleting it. (By the way, “albeit” should read “although”). The conclusion that follows can stand without the qualifier.

13.  Line 73: I am not aware of any evidence that MEN1 is associated with parathyroid carcinomas. The references cited do not support this either. The authors need to explain and justify this statement, or delete it (Actually, I don’t think it can be justified, so I recommend to delete it).

14.  Line 74: The evidence on F-choline PET/CT should also be discussed.

15.  Line 75: The ultrasound aspects are only suggestive, not diagnostic. Please qualify using “ultrasonographic features suggesting possible malignancy”.

16.  Line 76: “is the” should read “this”.

17.  Line 76: Please add, “except for the most superficial ones” at the end of the sentence. Part of the upper mediastinum is actually accessible to US examination, especially with appropriate placement of the patient’s neck and head, as well as by asking the patient to swallow.

18.  Line 76-80: When referring to your own work and previous studies, please say “we”, not “Schulte et al.” or “they”.

19.  Line 78: Strictly “orthotopic” or “cervical” in general?

20.  Line 86: “to characterize mediastinal parathyroid cancer as an entity”. Please delete this, it is not part of the study. The rest of the sentence described accurately the purpose and scope of the study (but please qualify using “recognition of POTENTIAL malignancy”, since most lesions that meet the 3x3 criterion are actually benign according to the data you report.

21.  Line 109: The search was performed in 2018 (line 97). How was it then possible to include “articles published between 1990-2020”? Also, since we are nearing the end of 2022, the authors need to update their literature search and data analysis accordingly.

22.  Lines 116-117: Please indicate in the text how many cervical PC and cervical PA there were.

23.  Lines 123-4: Since “AND” is in capital letters, “or” should be as well. Actually, the authors should delete Lines 124-125 because they are inaccurate and create confusion: even lesions that do meet the “3+3 rule” are “likely benign”, since the risk of malignancy is up to 10% (line 81).

24.  Ethics statements: “Institutional Review Board Statement: Not applicable. “Informed Consent Statement: Patient consent was waived as patient data was retrieved from prior published studies.” Who waived the consent if not the IRB?

25. Also about the ethics: In lines 94-95, the authors state that “We included 34 patients with MPA and one with MPC who had been operated at our center in the analysis.”. This is in direct contradiction to the Informed Consent Statement quoted above...

Author Response

Dear Reviewer,

we are greatly indebted for your careful and highly knowledgeable review of our draft manuscript. We have made all the proposed changes and believe that the manuscript is much better for it.  Please see attached word file.  

All best wishes

Reviewer 3 Report

This retrospective study evaluates the feasibility of application the 3+3 rules (tumor size ≥3 cm; calcium level ≥3 mmol/L) in mediastinal parathyroid cancer. The authors found that this criterion has a sensitivity of 73.9% and specificity of 84.1% to predict malignancy. One-third of the patients who is eligible for this criterion would turn out to be cancer. The inverse version, namely, “the negative 3+3 rules” validated that the mediastinal parathyroid cancer did not occur in tumor less than 3 cm with serum calcium level less than 3 mmol/L. The analysis is comprehensive and convincing, and the proposed criteria is practical. Congratulation for your great work.  

Please address the following concerns.

1.      Line 150: 40.0% of patients with reported follow-up died of cancer. Can you tell us how do you calculate the percentage?

Based on the number on Table 1, among 21 patients with eligible data, 6 patients dead of disease (DOD), and the rest 15 patients did not DOD. I think the percentage should be 6/ (6+15) =28.6%, rather than 6/15=40%, please confirm.

2.      Table 1,

a)      left column, recurrent cancer after 1st surgery excludes “pat” with persistence. “pat” is the abbreviation of patient or typographical error?

b)      left column, “delay to recurrence”.  Would it be better to use “time to recurrence”

Author Response

Dear Reviewer,

thank you for your careful review of our manuscript. We have made the proposed changes - please see the attached file.

Thank you
